# ALPBench: A Benchmark for Active Learning Pipelines on Tabular Data

## Abstract

In settings where only a budgeted amount of labeled data can be afforded, active learning seeks to devise query strategies for selecting the most informative data points to be labeled, aiming to enhance learning algorithms' efficiency and performance. Numerous such query strategies have been proposed and compared in the active learning literature. However, the community still lacks standardized benchmarks for comparing the performance of different query strategies. This particularly holds for the combination of query strategies with different learning algorithms into active learning pipelines and examining the impact of the learning algorithm choice. To close this gap, we propose `ALPBench`, which facilitates the specification, execution, and performance monitoring of active learning pipelines. It has built-in measures to ensure evaluations are done reproducibly, saving exact dataset splits and hyperparameter settings of used algorithms. In total, `ALPBench` consists of 86 real-world tabular classification datasets and 5 active learning settings, yielding 430 active learning problems. To demonstrate its usefulness and broad compatibility with various learning algorithms and query strategies, we conduct an exemplary study evaluating 9 query strategies paired with 8 learning algorithms in 2 different settings.

## 1 Introduction

Supervised learning requires labeled data, i.e., a collection of data points labeled with regard to the respective learning task. However, labeling data is usually time-consuming and expensive, e.g., if it has to be done by human domain experts (Settles et al., 2008). Collecting unlabeled data is often more affordable in terms of cost and easier to obtain, but not directly useful for supervised learning.

For situations where only a limited budget is available for labeling data, the field of active learning (AL) (Settles, 2009) develops methods for selecting the most suitable data points from unlabeled data to be labeled by a so-called *oracle*. The notion of "most suitable" here refers to data points that help achieve the best possible generalization performance for a given learning algorithm.

While AL is in principle applicable to different data modalities, such as images, text, video, or tabular data, each of these modalities presents unique challenges that affect not only the learning algorithm but also the active learning strategies (Werner et al., 2024). For instance, image data often involves high-dimensional, spatially correlated features, whereas tabular data requires handling mixed feature types, missing features, etc. (Shwartz-Ziv and Armon, 2022). In this work, we specifically focus on tabular data, which is widely used across various sectors, including medicine (Przystalski and Thanki, 2023), insurance (Hussain and Prieto, 2016), and manufacturing (Chen et al., 2023), and hence highly relevant for many real-world machine learning applications (Chui et al., 2018).

For tabular data, a diverse array of query strategies (QSs) are available in the literature that quantify the suitability of a data point in different ways, e.g., by Seung et al. (1992); Lewis and Gale (1994); Scheffer et al. (2001); Houlsby et al. (2011); Kirsch et al. (2021), to name a few. However, the performance of a QS depends on various factors, including the dataset, the budget constraints, and the learning algorithm, among other things (Evans et al., 2013; Ramirez-Loaiza et al., 2017; Pereira-Santos et al., 2019). Several empirical evaluations have already been conducted in the tabular data domain (Yang et al., 2018; Zhan et al., 2021; Bahri et al., 2022a; Lu et al., 2023). Still, the community lacks a benchmark for comparing the performances of different QSs that standardizes evaluation protocols and facilitates their comparison. Moreover, the existing evaluations are often limited in the

Figure 1: The contributions of our paper are the following: (i) the first active learning benchmark considering pipelines of query strategies and learning algorithms, (ii) an extensible Python package for applying and benchmarking active learning pipelines, and (iii) an extensive empirical evaluation of active learning pipelines.

number of datasets, considering only binary classification datasets or already outdated QSs (Yang et al., 2018; Lu et al., 2023). Further studies only consider one particular learning algorithm (Zhan et al., 2021; Bahri et al., 2022a; Lu et al., 2023), which can lead to biased results, as the algorithm also influences the performance of a QS (Ramirez-Loaiza et al., 2017). Lastly, these learning algorithms often do not properly represent state-of-the-art (SOTA) methods. For example, although gradient-boosted decision tree (GBDT) ensembles, such as XGBoost (Chen and Guestrin, 2016) or Catboost (Dorogush et al., 2018), as well as deep learning architectures (Arik and Pfister, 2021; Hollmann et al., 2023) have proven particularly successful for tabular data, they are not included in these studies.

**Contributions.** Thus far, a comprehensive benchmark to investigate the benefits of different query strategies in combination with different learning algorithms remains absent. Moreover, the field lacks a standardized evaluation framework to ensure fair comparisons and promote reproducible research. In this work, we address these gaps by proposing `ALPBench`, a comprehensive benchmark for active learning pipelines in the domain of tabular data classification tasks.

1. We propose `ALPBench`, the first tabular-data active learning benchmark that combines different learning algorithms and query strategies into active learning pipelines to execute and benchmark them against other pipelines across different settings and metrics.

2. We provide an implementation of `ALPBench` as an extensible Python package [1], offering standardized evaluation protocols to ensure consistent and reliable research outcomes. In an experimental study we showcase its usefulness by evaluating 72 different active learning pipelines on 86 real-world classification datasets across 2 settings and 2 metrics.

**Lessons learned.** In the following, we present a summary of our key findings, including insights into the performance differences between different learners, binary and multi-class datasets, different metrics and the scalability across small and large settings.

1. **Different learners:** We confirm that MarginSampling is a highly effective query strategy, particularly when combined with tree-based models. For models like SVM, KNN, and TabNet, representation-based approaches such as TypicalClustering prove to be better suited. FALCUN performs exceptionally well with MLPs.

2. **Different datasets:** For binary datasets, uncertainty-based methods combined with strong learners prove to be best, as these models provide reliable uncertainty estimates. However, as the number of classes increases, the data distribution might become more challenging to learn, and the benefit of incorporating representation or diversity-based approaches becomes more apparent.

3. **Different metrics:** When evaluating for accuracy, Margin Sampling is one of the most effective query strategies. For AUC, methods incorporating diversity, such as, e.g., ClusterMargin or PowerMargin, deliver the best performance. The importance of diversity might arise because achieving a high AUC requires a well-balanced representation of all classes in the dataset.

4. **Different settings:** The dominance of MarginSampling in the large data setting is reduced in the small setting, where methods that incorporate diversity excel. In these scenarios, having representative samples becomes more crucial, whereas in the large data setting, the initial samples may already provide sufficient information to cover different classes.

---

[1] https://anonymous.4open.science/r/alpbench-iclr25-F8E5/

## 2 RELATED WORK

Various active learning benchmarks have been proposed in the literature, each focusing on different domains such as image (Beck et al., 2021; Li et al., 2022; Zhang et al., 2023), text (Vysogorets and Gopal, 2024), or tabular data (Bahri et al., 2022a; Lu et al., 2023). Recently, Werner et al. (2024) explored active learning across multiple domains. However, their analysis is limited to linear models and deep neural networks. Considering that models like GBDTs and TabPFN excel on tabular data (McElfresh et al., 2023), and our aim to also investigate the interplay between learning algorithms and query strategies, we focus specifically on the tabular domain and integrate models tailored for this data type.

In the tabular data domain, an early benchmark of AL demonstrated that margin sampling (MS) often outperforms other QSs (Schein and Ungar, 2007) in combination with logistic regression (LR) as a learning algorithm. The performance of combining varying learning algorithms and QSs was investigated by Evans et al. (2013); Ramirez-Loaiza et al. (2017); Pereira-Santos et al. (2019). However, the studies are outdated, i.e., there are stronger machine learning (ML) algorithms nowadays (Grinsztajn et al., 2022; McElfresh et al., 2023), and many of the used datasets from the UCI repository (Newman and Merz, 1998) are rather old.

More recent QSs were investigated by Yang et al. (2018); Zhan et al. (2021); Lu et al. (2023). Although varying the strategy for instance selection, the learning algorithm is fixed, precisely a support vector machine (SVM) (Zhan et al., 2021; Lu et al., 2023) or LR (Yang et al., 2018). However, as the employed learner is crucial to the overall performance of AL (Ramirez-Loaiza et al., 2017), such design choice raises the question of whether the findings generalize to other learners as well. Further, their scope is limited to binary or only a handful of multi-class datasets.

All mentioned tabular benchmarks so far only considered one specific AL setting, i.e., the size of the initially labeled pool and the budget. Yang et al. (2018) initially provided only one labeled instance for each class, compared to, e.g., Lu et al. (2023), who randomly sampled 20 instances for the labeled pool. These misalignments across different benchmarks complicate comparisons and hinder the ability to draw general conclusions. Bahri et al. (2022a) were the first to address this issue by investigating three different AL settings. They also considered very recent QSs and datasets from the OpenML-CC18 Benchmark Suite (Bischl et al., 2019). However, again, the authors chose only a single specific learner, in this case, a deep neural network. Motivated by recent works by Grinsztajn et al. (2022) and McElfresh et al. (2023), we believe that an up-to-date benchmark has to include multiple SOTA learning algorithms for tabular data such as GBDTs (e.g., Catboost) and prior-fitted networks (PFNs) (e.g., TabPFN (Hollmann et al., 2023)) as well as recent QSs, e.g., power margin sampling and power BALD (Kirsch et al., 2021). To the best of our knowledge, we are the first to combine various SOTA learning algorithms with QSs and evaluate their performance on a large amount of binary and multi-class real-world classification tasks for tabular data. To address the challenges of evaluating active learning pipelines (Lüth et al., 2023), we provide standardized evaluation protocols across multiple settings, and metrics.

## 3 POOL-BASED ACTIVE LEARNING

In pool-based AL instances from the pool of unlabeled data are selected to be labeled by an oracle, which is done in an iterative procedure. Three different scenarios are commonly considered in AL, namely, the membership-query synthesis, the stream-based, and pool-based scenario (Settles, 2009; Tharwat and Schenck, 2023). We focus on the pool-based scenario, being the preferred one in real-world applications (Tharwat and Schenck, 2023). We first describe this scenario in Section 3.1 before elaborating on the implemented learning algorithms and QSs within `ALPBench` (Sections 3.2 and 3.3), and their combination into active learning pipelines (ALPs) (Section 3.4).

### 3.1 PROBLEM DEFINITION

In the classification setting, we are given a $d$-dimensional feature space $\mathcal{X} \in \mathbb{R}^d$ and a label set $\mathcal{Y} = \{1, ..., C\}$. A dataset (DS) is denoted as $\mathcal{D} = \{(\mathbf{x}_i, y_i)\}_{i=1}^n \subset \mathcal{X} \times \mathcal{Y}$, where each instance $\mathbf{x}_i = (x_i^1, \ldots, x_i^d) \in \mathcal{X}$ is associated with an underlying true label $y_i \in \mathcal{Y}$. In AL, however, only a small DS $\mathcal{D}_L^0 = \{(\mathbf{x}_i, y_i)\}_{i=1}^l$ is initially labeled, whereas a considerably larger pool of instances

$\mathcal{D}_U = \{(\mathbf{x}_i)\}_{i=l+1}^{n}$ is unlabeled. From this unlabeled pool, a QS selects instances to be labeled by the oracle $\mathcal{O}$. More specifically, the goal is to strategically select instances such that the predictive (probabilistic) model $h : \mathcal{X} \to \mathbb{P}(\mathcal{Y})$ induced by the learning algorithm on the labeled data minimizes the generalization error (risk) with respect to a given loss function $\ell : \mathcal{Y} \times \mathbb{P}(\mathcal{Y}) \to \mathbb{R}^+$. Here, $\mathbb{P}(\mathcal{Y})$ denotes the space of probability distributions over $\mathcal{Y}$. A given budget of $B$ can be spent for labeling, meaning that $B$ instances from $\mathcal{D}_U$ can be chosen and queried to $\mathcal{O}$. In the pool-based scenario, a predefined amount of $R$ instances is queried per iteration ($R \leq B$) and added to the current labeled DS $\mathcal{D}_L^i$, on which the learning algorithm is run to induce an updated model $h$.

## 3.2 LEARNING ALGORITHMS

The choice of the learning algorithm is quite important for the overall success of AL (Dos Santos and Carvalho, 2016). However, existing benchmarks typically fix a single learning algorithm, such as a deep neural network (DNN) (Bahri et al., 2022a) or an SVM (Lu et al., 2023), and recommend suitable QSs for this choice. To reveal insights for suitable QSs based on different learning algorithms, we investigate a variety of models. In particular, we choose the following models, covering a wide range of model types and including SOTA algorithms for tabular data (McElfresh et al., 2023): SVM, k-nearest neighbor (k-NN), random forest (RF), extremely randomized trees (ETC), LR, and naïve Bayes (NB) represent the group of base learners. For each of them, we implement multiple instantiations with different parameters. Further, we choose two GBDTs, namely XGBoost (XGB) and Catboost. Finally, we include a multi-layer perceptron (MLP) and TabNet (Arik and Pfister, 2021) as representatives of DNNs, and TabPFN (Hollmann et al., 2023) representing PFNs.

## 3.3 QUERY STRATEGIES

Query strategies (QSs) can be classified into information-based (Info.), representation-based (Repr.), and hybrid strategies (Hybr.) (Settles, 2009; Tharwat and Schenck, 2023). Info.-based strategies leverage the predictions of the learning algorithm to select instances where the learner exhibits uncertainty, as from these instances we expect the most informative insights. Repr.-based strategies rely solely on the structure of the data to identify the most representative instances. Hybr. strategies combine both of the aforementioned strategies.

Formally, let $z_i \in \mathcal{Z}$ either be a raw input instance, or its embedding of a neural network, $p_i \in \mathcal{P}$ the predicted class probabilities of a learning algorithm for that instance and $\{(\mathbf{x}_i)\}_{i=1}^{\mathcal{R}} \subseteq \mathcal{D}_U$ the pool of instances that is queried by the QS in each iteration. Loosely speaking, info.-based approaches select instances based on some uncertainty measure $u(\cdot)$ on the probability scores, repr.-based compute representativeness $r(\cdot)$ leveraging the structure of $\mathcal{Z}$; hybr. approaches combine both:

$$\{(\mathbf{x}_i)\}_{i=l+1}^{\mathcal{R}} \sim u(p_i) \qquad \{(\mathbf{x}_i)\}_{i=l+1}^{\mathcal{R}} \sim r(z_i) \qquad \{(\mathbf{x}_i)\}_{i=l+1}^{\mathcal{R}} \sim u(p_i) + r(z_i)$$

$$\text{Information-based} \qquad \text{Representation-based} \qquad \text{Hybrid}$$

*Information-based.* Information or uncertainty-based approaches calculate the uncertainty for each instance in the unlabeled pool, leveraging probability scores of the learning algorithm to subsequently select the most uncertain instances. These approaches are quite fast, as the calculations are performed in the (lower-dimensional) space of probabilities. However, they bear the risk of leading to a strong shift in the data distribution. We implement various approaches, most of which were also considered by Bahri et al. (2022a). Amongst them are the well-known margin sampling (MS) (Scheffer et al., 2001), entropy sampling (ES) (Shannon, 1948) and least-confident sampling (LC) (Lewis and Gale, 1994), sampling instances which have the lowest margin, highest entropy, or where the learning algorithm is the least-confident about, respectively. The QSs variance reduction (VR) (Cohn, 1993) and expected error reduction (EER) (Roy and McCallum, 2001) select instances that are expected to reduce the prediction error or output variance, respectively, and epistemic uncertainty sampling (EU) (Nguyen et al., 2019) samples instances, where the model exhibits uncertainty due to a lack of knowledge. Further, we also considered methods that compute uncertainty based on predicted probabilities of an ensemble such as query-by-committee (QBC) (Seung et al., 1992) (disagreement of the ensemble members), maximum entropy (MaxEnt) (Gal et al., 2017) (entropy of the averaged predictions) and BALD (Houlsby et al., 2011) (difference between MaxEnt and the averaged entropy

Table 1: Predefined active learning settings in `ALPBench`.

| | Static | | | Dynamic | |
|---|---|---|---|---|---|
| | small | medium | large | small | large |
| $\|D_L^0\|$ | 30 | 100 | 300 | $5 \cdot \|\mathcal{Y}\|$ | $20 \cdot \|\mathcal{Y}\|$ |
| $B$ | 200 | 1,000 | 4,000 | $100 \cdot \|\mathcal{Y}\|$ | $400 \cdot \|\mathcal{Y}\|$ |
| $R$ | 10 | 50 | 200 | $5 \cdot \|\mathcal{Y}\|$ | $20 \cdot \|\mathcal{Y}\|$ |

of the members' predictions). PowMS and PowBALD (Kirsch et al., 2021) build on MS and BALD but add a noise term to the uncertainty scores to enforce diversity within the queried instances.

*Representation-based.* QSs compute the representativeness of each instance in the raw input space or in some feature space. Both can potentially be high-dimensional, leading to high computational costs. K-means sampling (k-means) (Kang et al., 2004) performs clustering of the instances in $\mathcal{D}_U$ and selects those that are nearest to the cluster centers. Typical clustering (TypClu) (Hacohen et al., 2022) clusters all instances in $\mathcal{D}_L$ and $\mathcal{D}_U$ and then selects instances that lie in clusters in which no instance of $\mathcal{D}_L$ is located. CoreSet (Sener and Savarese, 2018) queries those instances from $\mathcal{D}_U$ for which the closest neighbor in $\mathcal{D}_L$ is the most distant.

*Hybrid.* Hybrid approaches combine uncertainty and representativeness. Cluster margin (CluMS) (Citovsky et al., 2021) selects instances by first performing clustering on $\mathcal{D}_U$ and then taking into account the margin scores as well. Clustering uncertainty-weighted embeddings (CLUE) (Prabhu et al., 2021) performs weighted k-means clustering on $\mathcal{D}_U$ with the entropy of the learning algorithm as sample weight. FALCUN (Gilhuber et al., 2024) computes a relevance score per instance, consisting of the margin scores of the learning algorithm and a diversity score.

### 3.4 ACTIVE LEARNING PIPELINES

We call the combination of a learning algorithm and a QS an active learning pipeline (ALP). Within an ALP, the learning algorithm and QS are used in alternating order to (re-)fit a model for the labeled data points and determine data points to be labeled by the oracle. In `ALPBench`, we explicitly account for this interplay and therefore allow for constructing ALPs out of every possible combination of learning algorithms and QS as long as they work with certain interfaces.

## 4 ACTIVE LEARNING PIPELINE BENCHMARK

`ALPBench` is meant to provide an easy-to-use and easy-to-extend platform for investigating ALPs, considering different combinations of learning algorithms and QSs, and evaluating new query strategies to be tested and compared against already known strategies. To this end, in `ALPBench`, we aim for high modularity with simple interfaces for the individual parts of an ALP, as well as for applying the composed pipelines to different datasets and experiment setups.

To facilitate the usage of `ALPBench`, we subsequently explain how AL problems and ALPs are specified, (Section 4.1 and 4.2, respectively), and what measures are taken for ensuring reproducibility and therewith high-quality experimental studies (Section 4.3).

### 4.1 SPECIFICATION OF ACTIVE LEARNING PROBLEMS

*Setting.* A setting describes the basic parameters of an AL benchmark problem. This includes the size of the test data and the initially labeled dataset, the number of AL iterations, and how many data points may be queried in each iteration. Acknowledging the impact of different sizes of the initial labeled pool $\mathcal{D}_L$ and the budget $\mathcal{B}$, we implemented three *static* settings, similar to Bahri et al. (2022a), and additionally two *dynamic* settings, as shown in Table 1. In the latter settings, the per-iteration budget is increasing with the number of classes in the dataset, as datasets with more classes are considered more challenging.

Table 2: Comparison of the scopes of `ALPBench` and previous benchmarks for tabular data.

| | | Yang et al. (2018) | Zhan et al. (2021) | Bahri et al. (2022a) | Lu et al. (2023) | **Ours** |
|---|---|---|---|---|---|---|
| QS | Info. | 8 | 7 | 8 | 6 | 13 |
| | Repr. | - | 2 | 2 | 2 | 3 |
| | Hybr. | 1 | 4 | 2 | 4 | 3 |
| Learner | Base | 1 | 1 | - | 1 | 6 |
| | GBDT | - | - | - | - | 2 |
| | DNN | - | - | 1 | - | 2 |
| | PFN | - | - | - | - | 1 |
| ALP | $\sum$ | 9 | 13 | 12 | 12 | **209** |
| DS | Binary | 44 | 35 | 35 | 26 | 48 |
| | Multi | - | 9 | 34 | - | 38 |
| | $\sum$ | 44 | 44 | 69 | 26 | **86** |
| AL Setting | | 1 | 1 | 3 | 1 | 5 |
| Metrics | | Accuracy | Accuracy | Accuracy | Accuracy | Accuracy, AUC, F1, Prec, Recall, Logloss |

*Scenario.* A scenario combines the fundamental parameters of a setting with a concrete classification task, i.e., an OpenML dataset ID, seeds for splitting the dataset into initially labeled, unlabeled, and test data, and a seed for pseudo-random execution of the active learning pipeline. By specifying a scenario, we, therefore, can describe a single active learning task. However, to conduct broader empirical studies, we need to have entire benchmark suites, which can also be specified in `ALPBench`.

*Benchmark Suite.* Benchmark Suites in `ALPBench` are essentially collections of datasets that can be combined with scenarios. `ALPBench` allows for specifying custom benchmark suites, with OpenML Feurer et al. (2021) serving as the backbone for datasets. To define new benchmark suites, it suffices to either give a benchmark ID from OpenML or specify a list of OpenML dataset IDs.

In our benchmark implementation, we provide five scenarios and two benchmark suites: OpenML-CC18 (Bischl et al., 2019) and TabZilla (McElfresh et al., 2023). Both benchmark suites together comprise a total of 86 datasets.

## 4.2 SPECIFICATION OF ACTIVE LEARNING PIPELINES

To apply AL methods to AL problems, active learning pipelines (ALPs) are specified by a learner and a query strategy (QS), as has been outlined in Section 3.4. They implement the main logic for the interplay between the learner and QS and take care of the communication with the oracle.

*Learner.* The learner is a learning algorithm that implements the scikit-learn classifier interface and is responsible for model induction. There are no restrictions on the type of learner as long as its interface matches that of a `scikit-learn` classifier. It is only provided with labeled data points.

*Query Strategy.* Provided with the learner, the already labeled and unlabeled data points, the QS selects unlabeled data points to be labeled by the oracle. While we wrap and include random sampling, BALD, QBC, EER and EU from the `scikit-activeml` library (Kottke et al., 2021), the remaining QSs are original implementations in `ALPBench`. In total, we include 19 QSs and a broad spectrum of 11 different learners that can be combined into more than 200 ALPs.

## 4.3 REPRODUCIBILITY AND EXPERIMENTATION

As we would like to ensure a high-quality standard for experiments conducted with `ALPBench`, we provide support for logging and facilitate the execution of experiments.

*Benchmark Connector.* The benchmark connector stores meta-information relevant for reproducibility. This includes storing the indices of data points that are labeled initially and used for testing. Furthermore, the settings of hyperparameters of learners and query strategies are stored so that the same configurations can be maintained for future studies. We provide two facades of the Benchmark Connector, one using a database as data storage and one that works locally with a filesystem.

*Experimenter.* Building on pyExperimenter (Tornede et al., 2023), `ALPBench` comes with some convenience functionalities to foster large-scale experimental studies. A cross-product experiment

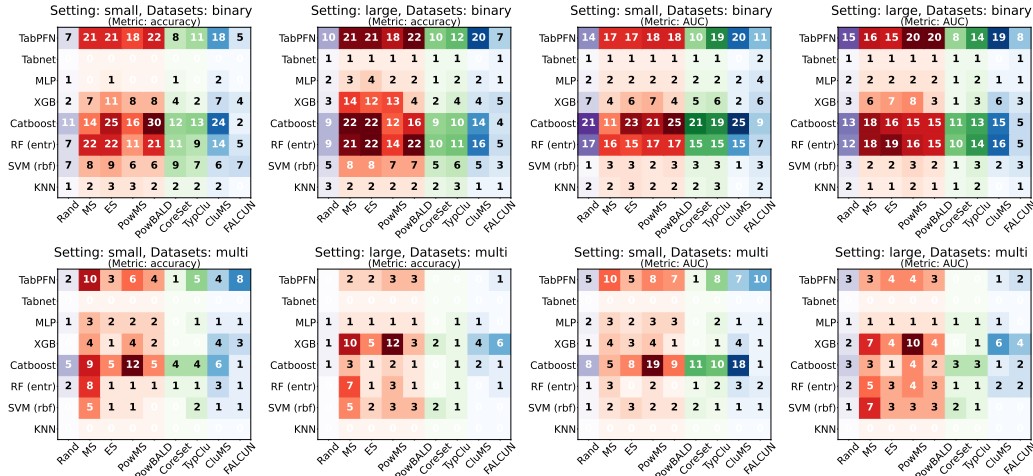

Figure 2: Heatmaps for all ALPs within our evaluation study using AUBC **(accuracy)** as performance measure (first and second column) and AUBC **(AUC)** (third and fourth), for **binary** (first row) and **multi-class** (second row) datasets. Information-based, representation-based, and hybrid QSs are colored in red, green, and blue, respectively, and random sampling in purple.

grid is specified for some default setup and can be easily extended by more alternatives. Furthermore, we provide logging facilities to observe the active learning process, recording labeling statistics and learner performances using different metrics.

In Table 2, we compare the scope of our benchmark to previous studies on active learning for tabular data (Yang et al., 2018; Zhan et al., 2021; Bahri et al., 2022a; Lu et al., 2023). Our work provides the most comprehensive benchmark so far, especially regarding the different chosen learning algorithms, settings, and metrics to be evaluated.

## 5 EXPERIMENTS

To demonstrate the usefulness of `ALPBench`, we conduct an empirical study comparing various active learning pipelines composed of different combinations of QSs and learning algorithms. We would like to emphasize that due to a large number of datasets and resulting ALPs, this (only) includes a carefully selected subset of the QSs and learning algorithms available within `ALPBench`. Concretely, we investigate the effectiveness of 9 QSs and pair them with 8 learning algorithms, constituting the most extensive study on active learning pipelines. The experimental setup is explained in Section 5.1 before the evaluation methods and results are described in Sections 5.2 and 5.3, respectively.

### 5.1 EXPERIMENTAL SETUP

In our experimental study, we select from the 19 QSs that `ALPBench` provides a set of 9 representative QS, covering the different types of query strategies. We also choose a subset of 8 learning algorithms from different ends of the bias-variance spectrum, ranging from linear to highly non-linear models, including various decision tree ensembles and SOTA deep learning methods for tabular data. More precisely, we include ES (Shannon, 1948), MS (Scheffer et al., 2001), PowMS and PowBALD (Kirsch et al., 2021), CoreSet (Sener and Savarese, 2018), FALCUN (Gilhuber et al., 2024), CluMS (Citovsky et al., 2021) and TypClu (Hacohen et al., 2022), and random sampling (Rand) as QSs and SVM, k-NN, MLP, RF, XGB, Catboost, TabNet, and TabPFN as learning algorithms.

*Datasets.* We evaluate each ALP on all DSs from the OpenML-CC18 (Bischl et al., 2019) and the TabZilla (McElfresh et al., 2023) benchmark suites, except for 4 quite large datasets. Precisely, we exclude the datasets with OpenML IDs 1567, 1169, 41147, and 1493, leaving us with 48 binary and 38 multi-class real-world datasets. The datasets from the TabZilla suite are found to be very challenging by the authors, and we anticipate they will similarly present challenges for AL.

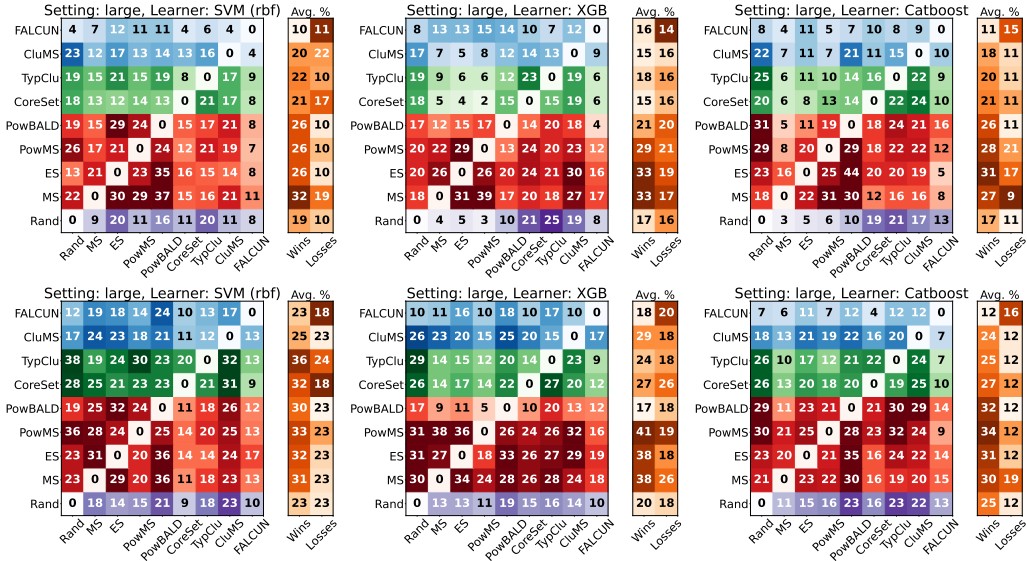

Figure 3: Win-Matrices for SVM, XGB and Catboost for the **large** setting using AUBC **(accuracy)** as performance measure (first row) and AUBC **(AUC)** (second row). The last columns in each figure show the average win and loss percentages.

*Settings.* We evaluate on the two *dynamic* settings (cf. Table 1), as we want to scale the budget with the task complexity, which increases with the number of classes. Further elaboration on the experimental setup, the configuration of the learning algorithms, and the hardware infrastructure is given in Appendix A.3.

## 5.2 EVALUATION METHODS

We firstly aim to investigate the interplay of the QS with different learners, to reveal which QSs are particularly effective for each learner. Adhering to common evaluation procedures for comparing QSs (Bahri et al., 2022b;a; Lu et al., 2023), we compute budget curves and win-matrices.

*Budget curves.* Budget curves quantify the (test) performance of an ALP at each round of the AL procedure. The area under the budget curve (AUBC) then offers a robust metric to compare different ALPs over the whole AL procedure, given this (test) performance. Within our benchmark, we tracked six different performance measures, as shown in Table 2, but in this evaluation study focused on accuracy and AUC, as they are most widely used in the AL literature (Ramirez-Loaiza et al., 2017). We denote the AUBC given both metrics as AUBC *(accuracy)* and AUBC *(AUC)*, respectively.

*Win-matrices.* For each learning algorithm, we compute a win-matrix $W$ to compare the performances of different QSs. Let $D$ be the number of available datasets and assume $M$ different QSs, this results in a matrix of size $M \times M$. To make the plots visually more appealing, we slightly modify the definition of the entry of $W$ at position $(i, j)$ compared to Bahri et al. (2022b;a) as follows

$$W_{(i,j)} = \sum_{d=1}^{D} \mathbb{1}[\text{QS i beats QS j on dataset d}].$$

To determine a win, we compare the AUBC of two QSs after the total amount of iterations. This provides us with a robust measure since the overall performance across all iterations is captured. Wins are only defined in case of statistical significance, using Welch's t-test with $p = 0.05$.

We further want to investigate whether strong learning algorithms for tabular data found by McElfresh et al. (2023) perform well in the low-label regime, especially when combined with QSs into ALPs.

*Heatmaps.* Sticking to the notation above and further assuming $N$ learning algorithms, we compute heatmaps $H$ of size $N \times M$. Let learner $i$ and QS $j$ form the combined $\text{ALP}_{(i,j)}$ and $\text{ALP}_d$ be the winning ALP for the dataset $d$, meaning it has the highest AUBC. Then, the entry of the heatmap at

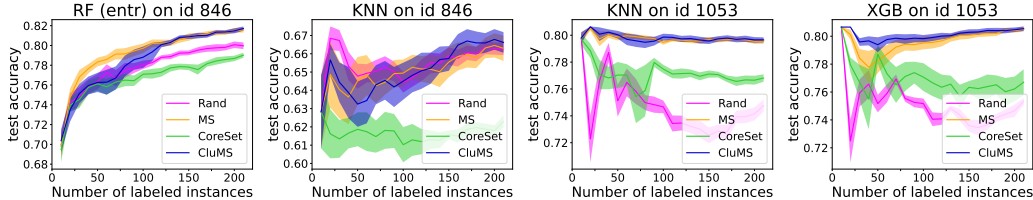

Figure 4: Budget curves for different ALPs combined of RF, k-NN, XGB and Rand, MS, CoreSet and CluMS on different datasets, considering the **small** setting.

position $(i, j)$ is defined as

$$H_{(i,j)} = \sum_{d=1}^{D} \mathbb{1}[\text{ALP}_{(i,j)} \text{ is not statistically significant from ALP}_d \text{ on dataset d}].$$

Statistical significance is determined similar as for the win-matrices, the indicator function now evaluates to one for $\text{ALP}_d$ and all ALPs for which the null hypothesis cannot be rejected.

## 5.3    RESULTS

In this section, we present our main insights, aiming to answer the following research questions (RQ):

*RQ1:* Which ALPs perform best and worst?

*RQ2:* Given a specific choice of the learning algorithm, setting, metric and types of datasets, which QS is particularly well suited?

*RQ3:* Are there datasets and/or settings where AL leads to a decrease in performance?

*RQ1.* In Figure 2, we show heatmaps as described in Section 5.2 evaluated on AUBC (accuracy) and AUBC (AUC), separately for binary and multi-class datasets and for the small and large setting.

RF, Catboost, and TabPFN are quite dominant, as they constitute to many winning ALPs, especially for the binary datasets. XGB also performs well overall, however, showing a preference for large settings and multi-class datasets. TabNet, MLP, and k-NN are performing inferior, which, in the case of TabNet, might be due to limited training time. Overall, information-based strategies are quite dominant, especially for binary datasets regarding AUBC (accuracy). We hence confirm the finding that MS is a very competitive QS if evaluated for AUBC (accuracy) (Schein and Ungar, 2007; Bahri et al., 2022a) and extend it to other learners. However, when the AUBC (AUC) is considered, QSs that incorporate also diversity are superior, especially for binary datasets. This particularly holds for CluMS and TypClu. Also Rand is more competitive in this scenario, which extends findings of Lu et al. (2023) for learners beyond a SVM. The QSs MS and power-set margin sampling (PowMS) are quite strong for multi-class datasets regarding both metrics. Moreover, the learning algorithm seems to be the crucial choice for the pipeline to achieve good performance.

*RQ2.* In Figure 3, we present win-matrices for different learning algorithms in the large setting. We choose SVM since it has been chosen as a learning algorithm in other AL studies, such as in Zhan et al. (2021); Lu et al. (2023). Further, we choose Catboost and XGB, as they have shown strong performance when combined into ALPs. We evaluate on all datasets and present results for the AUBC (accuracy) in the first row and AUBC (AUC) and in the second row, where the last columns in each Figure indicate the average win and loss percentages. Further win-matrices are provided in the Appendix A.4.

The win-matrices clearly show that the suitability of different QSs varies, depending on the given learning algorithm and metric. Regarding the AUBC (accuracy), MS is quite dominant, if the learner is chosen to be Catboost or XGB. This can be also deduced from the high win percentage. If the learner is an SVM, however, all other information-based QS and also TypClu outperform MS with respect to the win to loose percentage ratio. For the AUBC (AUC), PowMS is very strong when combined with Catboost and XGB. Regarding the SVM, both representation-based QSs outperform all other information-based strategies.

*RQ3.* In Figure 4, we present budget curves for RF, k-NN, and XGB on two datasets (OpenML ID 846 and 1053) in the small setting. For better visual clarity, we only combine the learners with Rand, MS, CoreSet, and CluMS, each representing a different type of QSs.

Budget curves in AL are generally expected to show an upward trend, indicating improved performance with an increasing budget, as visual in the first subfigure. However, this pattern is not consistent across all learners, as for the combination of k-NN and CoreSet, the performance decreases. On a different dataset (third subfigure), this also holds for random sampling and slightly for MS and CluMS. Even for a learning algorithm that demonstrates a strong overall performance, the picture can look quite similar, as shown in the fourth subfigure. To conclude, AL can deteriorate performance, as has also been shown by Guo and Schuurmans (2007a) and Gasperin (2009). We again want to emphasize the strong dependence of the performance of ALPs on the chosen learning algorithm, dataset, setting, and potentially other properties, which still need to be understood. We hope that `ALPBench` will serve as a tool to gain more insights into this.

## 6   CONCLUSION AND FUTURE WORK

We proposed `ALPBench`, a benchmark for active learning pipelines (ALPs) on tabular data. `ALPBench` allows for easily combining QSs and learning algorithms into ALPs and provides a unified API to evaluate and benchmark them against each other. The open-source implementation of our benchmark is available as a Python package.

In the benchmark so far, we predefined five different settings, which were partly inspired by Bahri et al. (2022a). However, the exploration of more settings having different requirements for suitable pairs of QSs and learning algorithms outlines an interesting avenue for future work. Further, it might be appealing to incorporate other more recent trends, such as label noise, multiple annotators, etc.

In our experimental evaluation, we find that most of the time, strong pipelines consist of learners such as RF, Catboost, or TabPFN and information-based query strategies. However, there is no clear SOTA QS, as the suitability of a QS heavily depends on the chosen learner, the metric to be evaluated and the specific dataset. For instance, we confirm that MS is highly competitive regarding the AUBC (accuracy) (Schein and Ungar, 2007; Bahri et al., 2022a) and extend this finding to learners like e.g., Catboost and XGB. However, when evaluating for AUBC (AUC), query strategies that also incorporate diversity tend to perform better. Additionally, for learners such as SVM or k-NN, representation-based approaches are more suitable.

With this benchmark and library, we hope to foster further research to fairly evaluate new QS considering different datasets, settings, and learners. Moreover, it might be appealing to specifically develop new QSs for certain settings and/or learners. Lastly, we would also like to study whether it might be advantageous to devise hyperheuristics switching between different QSs within one active learning procedure.

## 7   LIMITATIONS AND BROADER IMPACT STATEMENT

Both the benchmark and the evaluation study are limited to tabular classification problems and consider a specific set of active learning settings. Furthermore, in the empirical study, we restricted the training time to 180 seconds per iteration, which might limit generalizability for the large settings. Nevertheless, we observe complementary performance for both learning algorithms and query strategies, which underpins the need for a benchmark like `ALPBench`.

Current active learning research often lacks consistency, as researchers choose different settings and learners to demonstrate the effectiveness of their proposed QS. However, key factors such as, e.g., the choice of learner, batch size, and number of iterations can significantly impact performance. Therefore, establishing a standardized evaluation framework is needed to ensure fair comparisons and encourage more consistent and comparable research in this area. `ALPBench` aims to serve as a starting point to address this need.

As this paper presents work that aims to advance the field of machine learning, there are many potential societal consequences of our work. However, we feel that none of these needs to be specifically highlighted here.

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

# A  APPENDIX

## A.1  GLOSSARY OF ACRONYMS

**AL**  active learning
**ALP**  active learning pipeline
**AUBC**  area under the budget curve
**DS**  dataset
**ML**  machine learning
**QS**  query strategy
**SOTA**  state-of-the-art
**DNN**  deep neural network
**ETC**  extremely randomized trees
**GBDT**  gradient-boosted decision tree
**k-NN**  k-nearest neighbor
**LR**  logistic regression
**MLP**  multi-layer perceptron
**NB**  naïve Bayes
**PFN**  prior-fitted network
**RF**  random forest
**SVM**  support vector machine
**XGB**  XGBoost
**AAL**  adaptive active learning
**ALBL**  active learning by learning
**BALD**  Bayesian active learning by disagreement
**CER**  combined error reduction
**CLUE**  clustering uncertainty-weighted embeddings
**CluMS**  cluster margin
**CoreSet**  CoreSet
**DWUS**  density weighted uncertainty sampling
**EER**  expected error reduction
**EMC**  expected model change
**ES**  entropy sampling
**EU**  epistemic uncertainty sampling
**EVR**  expected variance reduction
**FALCUN**  fast active learning by contrastive uncertainty
**FIVR**  Fisher information variance reduction
**GRAPH**  graph density
**HIER**  hierarchical sampling
**LC**  least-confident sampling
**k-means**  k-means sampling
**MarginDensity**  pre-clustering and margin sampling
**MaxEnt**  maximum entropy
**MaxER**  maximum error reduction
**MinMS**  minimum margin sampling
**MLI**  minimum loss increase
**MMC**  maximum model change
**MS**  margin sampling
**PowBALD**  power-set BALD
**PowMS**  power-set margin sampling
**QBC**  query-by-committee
**QBC VR**  QBC VR
**QUIRE**  querying informative and representative examples
**Rand**  random sampling
**TypClu**  typical clustering
**VR**  variance reduction

## A.2 COMPARISON TO EXISTING BENCHMARKS FOR TABULAR DATA

In the following, we present an extensive table which compares `ALPBench` with existing active learning benchmarks. The QS and learning algorithms are ordered by their year of appearance. In Table 3, we additionally present a detailed version of Table 2 in the main paper, which shows which exact QS and learners were implemented in the benchmarks.

| Query Strategy | Year | Yang et al. (2018) | Zhan et al. (2021) | Bahri et al. (2022a) | Lu et al. (2023) | ALPBench |
|---|---|---|---|---|---|---|
| ES Shannon (1948) | 1948 | ✓ | ✓ | ✓ | ✓ | ✓ |
| QBC Seung et al. (1992) | 1992 | ✗ | ✓ | ✗ | ✓ | ✓ |
| VR Cohn (1993) | 1993 | ✗ | ✓ | ✗ | ✓ | ✓ |
| LC Lewis and Gale (1994) | 1994 | ✗ | ✓ | ✓ | ✓ | ✓ |
| FIVR Zhang (2000) | 2000 | ✓ | ✗ | ✗ | ✗ | ✗ |
| MS Scheffer et al. (2001) | 2001 | ✗ | ✓ | ✓ | ✓ | ✓ |
| EER Roy and McCallum (2001) | 2001 | ✓ | ✓ | ✗ | ✓ | ✓ |
| MaxER Guo and Greiner (2007) | 2007 | ✓ | ✗ | ✗ | ✗ | ✗ |
| CER Guo and Schuurmans (2007b) | 2007b | ✓ | ✗ | ✗ | ✗ | ✗ |
| EVR Schein and Ungar (2007) | 2007 | ✓ | ✗ | ✗ | ✗ | ✗ |
| EMC Settles et al. (2007) | 2007 | ✗ | ✓ | ✗ | ✗ | ✗ |
| MLI Hoi et al. (2008) | 2008 | ✓ | ✗ | ✗ | ✗ | ✗ |
| BALD Houlsby et al. (2011) | 2011 | ✗ | ✗ | ✓ | ✗ | ✓ |
| MMC Cai et al. (2017) | 2017 | ✓ | ✗ | ✗ | ✗ | ✗ |
| MaxEnt Gal et al. (2017) | 2017 | ✗ | ✗ | ✓ | ✗ | ✓ |
| QBC VR Beluch et al. (2018) | 2018 | ✗ | ✗ | ✓ | ✗ | ✓ |
| EU Nguyen et al. (2019) | 2019 | ✗ | ✗ | ✗ | ✗ | ✓ |
| PowMS Kirsch et al. (2021) | 2021 | ✗ | ✗ | ✓ | ✗ | ✓ |
| MinMS Jiang and Gupta (2021) | 2021 | ✗ | ✗ | ✓ | ✗ | ✓ |
| k-means Kang et al. (2004) | 2004 | ✗ | ✓ | ✗ | ✗ | ✓ |
| HIER Dasgupta and Hsu (2008) | 2008 | ✗ | ✓ | ✗ | ✓ | ✗ |
| CoreSet Sener and Savarese (2018) | 2018 | ✗ | ✗ | ✓ | ✓ | ✓ |
| TypClu Hacohen et al. (2022) | 2022 | ✗ | ✗ | ✓ | ✗ | ✓ |
| MarginDensity Nguyen and Smeulders (2004) | 2004 | ✗ | ✗ | ✓ | ✗ | ✗ |
| DWUS Settles and Craven (2008) | 2008 | ✗ | ✓ | ✗ | ✓ | ✗ |
| QUIRE Huang et al. (2010) | 2010 | ✗ | ✓ | ✗ | ✓ | ✗ |
| GRAPH Ebert et al. (2012) | 2012 | ✗ | ✓ | ✗ | ✓ | ✗ |
| AAL Li and Guo (2013) | 2013 | ✓ | ✗ | ✗ | ✗ | ✗ |
| ALBL Hsu and Lin (2015) | 2015 | ✗ | ✓ | ✗ | ✓ | ✗ |
| CluMS Citovsky et al. (2021) | 2021 | ✗ | ✗ | ✓ | ✗ | ✓ |
| CLUE Prabhu et al. (2021) | 2021 | ✗ | ✗ | ✗ | ✗ | ✓ |
| FALCUN Gilhuber et al. (2024) | 2024 | ✗ | ✗ | ✗ | ✗ | ✓ |

| Learning Algorithm | Year | Yang et al. (2018) | Zhan et al. (2021) | Bahri et al. (2022a) | Lu et al. (2023) | ALPBench |
|---|---|---|---|---|---|---|
| LR Berkson (1944) | 1944 | ✓ | ✗ | ✗ | ✗ | ✓ |
| k-NN Fix and Hodges (1952) | 1952 | ✗ | ✗ | ✗ | ✗ | ✓ |
| MLP Werbos (1974) | 1974 | ✗ | ✗ | ✓ | ✗ | ✓ |
| NB Kononenko (1990) | 1990 | ✗ | ✗ | ✗ | ✗ | ✓ |
| SVM Boser et al. (1992) | 1992 | ✗ | ✓ | ✗ | ✓ | ✓ |
| RF Breiman (2001) | 2001 | ✗ | ✗ | ✗ | ✗ | ✓ |
| ETC Geurts et al. (2006) | 2006 | ✗ | ✗ | ✗ | ✗ | ✓ |
| XGB Chen and Guestrin (2016) | 2016 | ✗ | ✗ | ✗ | ✗ | ✓ |
| Catboost Dorogush et al. (2018) | 2018 | ✗ | ✗ | ✗ | ✗ | ✓ |
| TabNet Arik and Pfister (2021) | 2021 | ✗ | ✗ | ✗ | ✗ | ✓ |
| TabPFN Hollmann et al. (2023) | 2023 | ✗ | ✗ | ✗ | ✗ | ✓ |

Table 3: Comparison of the scopes of `ALPBench` and previous benchmarks for tabular data.

| | | Yang et al. (2018) | Zhan et al. (2021) | Bahri et al. (2022a) | Lu et al. (2023) | **Ours** |
|---|---|---|---|---|---|---|
| QS | Info. | ES, MaxER, MMC, FIVR, EER, CER, EVR, MLI | ES, QBC, VR, LC, MS, EER, EVR | ES, LC, MS, BALD, MaxEnt, QBC VR, PowMS, MinMS, PowBALD | ES, QBC, VR, LC, MS, EER | ES, QBC, VR, LC, MS, EER, BALD, MaxEnt, QBC VR, EU, PowMS, MinMS, PowBALD |
| | Repr. | - | k-means, HIER | CoreSet, TypClu | HIER, CoreSet | k-means, CoreSet, TypClu |
| | Hybr. | AAL | DWUS, QUIRE, GRAPH, ALBL | MarginDensity, CluMS | DWUS, QUIRE, GRAPH, ALBL | CluMS, CLUE, FALCUN |
| Learner | Base | LR | SVM | - | SVM | k-NN, SVM, RF, LR, NB, ETC |
| | GBDT | - | - | - | - | CatBoost, XGB |
| | DNN | - | - | MLP | - | MLP, TabNet |
| | PFN | - | - | - | - | TabPFN |
| ALP | $\sum$ | 9 | 13 | 12 | 12 | **209** |
| DS | Binary | 44 | 35 | 35 | 26 | 48 |
| | Multi | - | 9 | 34 | - | 38 |
| | $\sum$ | 44 | 44 | 69 | 26 | **86** |
| AL Setting | | 1 | 1 | 3 | 1 | 5 |
| Metrics | | Accuracy | Accuracy | Accuracy | Accuracy | Accuracy, AUC, F1, Prec, Recall, Logloss |

## A.3 EXPERIMENTS

In this section, we elaborate in more detail on the experiments that were conducted within our evaluation study.

*Datasets.* From the 90 datasets from the OpenML-CC18 Bischl et al. (2019) and the TabZilla Benchmark Suite McElfresh et al. (2023) we filtered and excluded the datasets with OpenML IDs 1567, 1169, 41147, and 1493. The first three were filtered out for all settings because they consist of more than 300,000 data points, which would result in a large amount of computing time for the non-info. based QSs. The last dataset with OpenML ID 1493 was filtered out since it consists of 100 classes, which would result in a huge amount of the per iteration budget $\mathcal{R}$, limiting the number of iterations to a high degree. Further, for the large setting, we wanted to guarantee that at least 10 iterations can be performed until all instances from $\mathcal{D}_U$ are queried. This led to the removal of OpenML IDs 11, 12, 14, 16, 18, 22, 25, 51, 54, 188, 307, 458, 469, 1468, 1501, 40966, and 40979 for this setting. For the preprocessing steps, we proceed as follows. Categorical features are one-hot encoded and missing values are imputed by the mean or mode of the corresponding feature.

*Active Learning Setting.* As mentioned, we investigate a small and a large setting. Explicitly, the small and large settings are specified by $|\mathcal{D}_L^0| = R = 5 \cdot |\mathcal{Y}|$ and $|\mathcal{D}_L^0| = R = 20 \cdot |\mathcal{Y}|$, respectively, for the given dataset and a total amount of 20 iterations or until all instances from the unlabeled pool $\mathcal{D}_U$ are queried. We choose the factor 5 for the small setting, since then $\mathcal{R}$ matches the one in the (static) small setting in Bahri et al. (2022a). For the large setting, we should have chosen a factor of 100 to be again consistent with Bahri et al. (2022a). However, this seemed unrealistic to us for real-world applications. For some (imbalanced) datasets, it may happen that not every class is at least once represented in $\mathcal{D}_L^0$. In these cases, we additionally randomly sample one instance from $\mathcal{D}_U$ per missing class and add them with their corresponding label to $\mathcal{D}_L^0$. We run each ALP ten times with different seeds, where the seed defines the $\frac{2}{3}/\frac{1}{3}$-split of the total dataset $\mathcal{D}$ into $\mathcal{D}_{\text{train}}$ and $\mathcal{D}_{\text{test}}$ as well as the split of $\mathcal{D}_{\text{train}}$ into $\mathcal{D}_L$ and $\mathcal{D}_U$. Needless to say, the datasets we consider are originally (fully-)labeled datasets. Tailored to the AL setting, we discard the labels for the instances in $\mathcal{D}_U$ and assure that only the oracle $\mathcal{O}$ can access them.

*Configuration of Learning Algorithms.* In general, we do not perform any hyperparameter optimization (HPO) but rather stick to the default parameters. To contain computational costs, we limit the training time of the learning algorithms. For XGB and Catboost, we reduce the training time by setting the tree method to *hist* and limiting the amount of iterations, respectively. For Catboost and for TabNet, we implement a timeout of three minutes per iteration for the same purpose. This of course may decrease the performance of the learning algorithms and poses a limitation to the generalizability of our empirical study. Further, TabPFN (Hollmann et al., 2023) can so far only be fitted on a maximum amount of 1,000 instances. Therefore, we uniformly sample 1,000 instances from the current dataset to be fitted on, in case this constraint is violated, similar to McElfresh et al. (2023). For TabPFN and TabNet we modify the implementation for the representation-based and hybrid approaches. Concretely, we extract the output of the encoder from the TabPFN and the activations of the

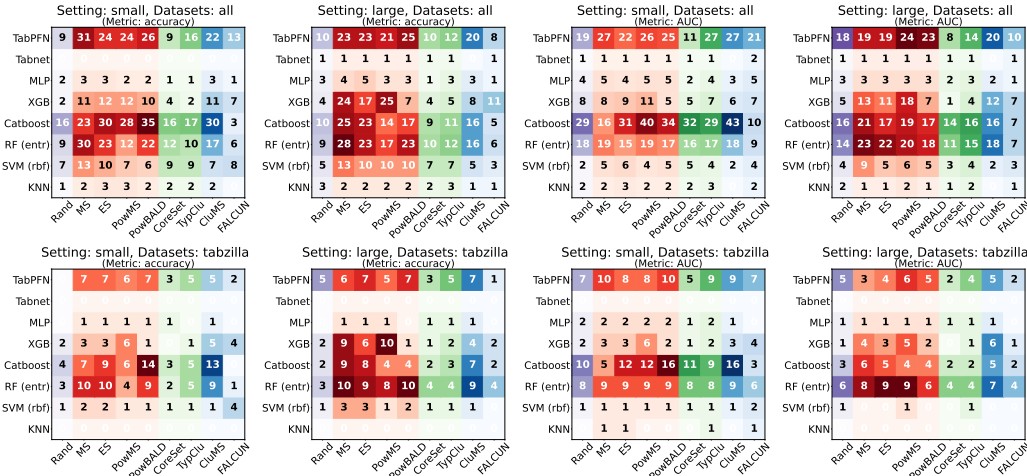

Figure 5: Heatmaps for all ALPs within our evaluation study using AUBC (**accuracy**) as performance measure (first and second column) and AUBC (**AUC**) (third and fourth), separately for **all** (first row) datasets and for the **TabZilla** (second row) datasets. Information-based, representation-based, and hybrid QSs are colored in red, green, and blue, respectively, and random sampling is in purple.

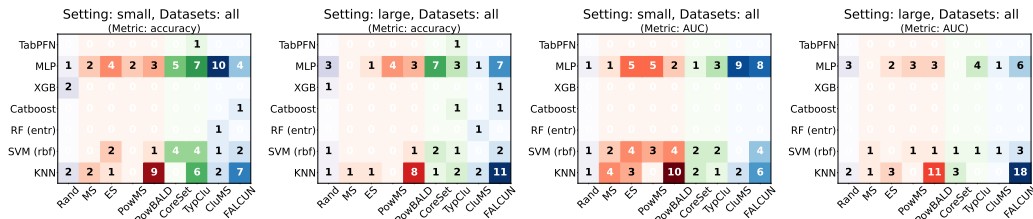

Figure 6: Lose-Heatmaps for all ALPs within our evaluation study using AUBC (**accuracy**) as performance measure (first and second subfigure) and AUBC (**AUC**) (third and fourth) on all datasets **without** statistical significance. The color-coding is consistent with Figure 5.

penultimate layer from TabNet to compute the representativeness of each instance based on its embedding. The exact details can obviously be looked up in our implementation.

*Implementation.* All experiments were conducted with 2 CPU cores and 8GiB RAM or 16GiB for the small and large settings, respectively, to resemble end-user environments. The HPC nodes for the computations are equipped with two AMD Milan 7763 and 256GiB main memory in total. Runs exceeding these limits have been canceled by the workload manager.

## A.4 RESULTS

This section contains more experimental results, comprising more heatmaps and win-matrices distinguishing between binary and multi-class datasets, small and large settings and different metrics. We also present more budget curves for other datasets and learners.

Precisely, we first present heatmaps where we - similar to the main paper - distinguish between small and large settings as well as both metrics AUBC (accuracy) and AUBC (AUC). However, we now compute heatmaps for all datasets (binary and multi-class combined) and for all datasets from the TabZilla Benchmark Suite McElfresh et al. (2023), cf. Figure 5 the first and the second row, respectively. The latter one is a selection of particulary hard or difficult datasets, so we suppose them to be hard for active learning as well.

The main trend of the results of all datasets looks quite similar to the binary datasets in the main paper: Most winning pipelines constitute of TabPFN, Catboost, XGB or RF as learner and information-based QS. However, CluMS is also part of many winning pipelines, especially in the small setting and Rand is quite competitive when considering AUC. For the TabZilla datasets, TabPFN and XGB appear to be not that strong. The QS k-NN and Tabnet (almost) never constitute a winning pipeline and CluMS again is competitive regarding both metrics, especially in the small setting.

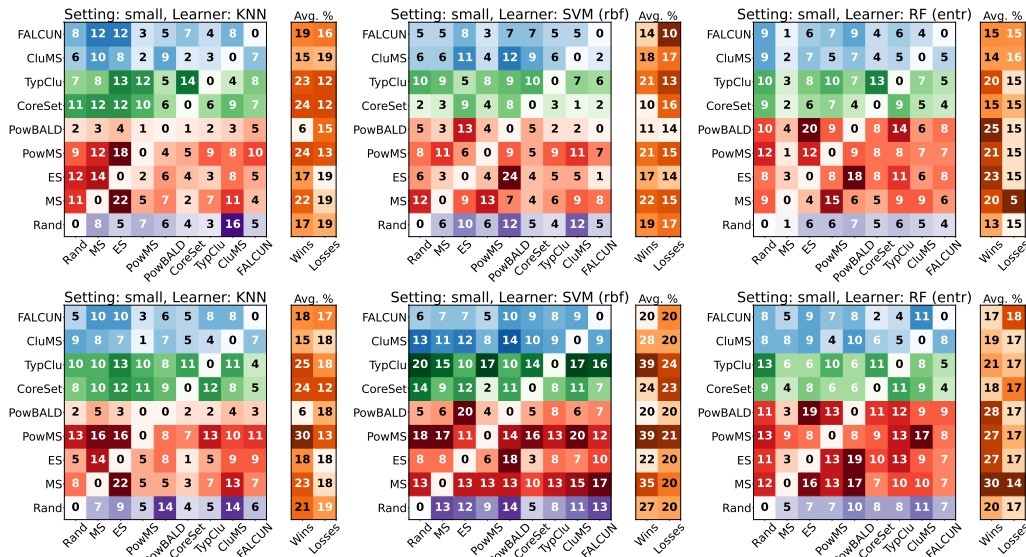

Figure 7: Win-Matrices for k-NN, SVM and RF for the **small** setting on **multi-class** datasets using AUBC **(accuracy)** as performance measure (first row) and AUBC **(AUC)** (second row).

To investigate which ALPs perform particularly poorly, we present *Lose-Heatmaps* in Figure 6, where the losing pipeline replaces $ALP_d$. Hereby, we do not separate between binary and multi-class datasets and further exclude TabNet as it did not perform at all in our investigated setting. We neglect statistical significance, which may seem an unusual perspective, but it helps to reveal insights into which ALPs exhibit the lowest performance for each dataset. In this figure, we find that it is more important to choose a strong learner than selecting a suitable QS. Concretely, one should avoid MLP or k-NN, and ALPs combining k-NN with PowBALD or MLP with FALCUN or CluMS proved disadvantageous. It might happen, that your learner is not strong, because you maybe want to use a very simple, interpretable model or the data is extremely difficult to learn. In this case it might not be a good idea to rely on any probabilistic estimates but rather choose Rand, as it rarely constitutes to loosing pipelines for k-NN and MLP.

In Figure 7 we present win-matrices for the learners k-NN, SVM and RF considering the small setting and evaluating on multi-class datasets. Hereby, we distinguish again between the metrics AUBC (accuracy) and AUBC (AUC). If the metric is chosen as accuracy, we make the following observations. For the k-NN the representation-based and hybrid approaches are very competitive with the information-based strategies. This effect decreases, when SVM is chosen and for the RF the information-based strategies are dominant with MS being extremely robust. In contrast to the RF, Rand is not a too bad choice for k-NN and SVM. Regarding the AUC, TypClu is quite strong for the SVM. For the RF, the information-based strategies are outperforming other QS and in particular MS is strong. Again, we see that the performance of all QSs depend on the chosen learning algorithm.

Further, we present budget curves comparing a subset of 5 different QS for enhanced visual clarity. Precisely, we chose Rand, two representatives for the information-based strategies (MS and power-set BALD (PowBALD)), and one representative for each remaining group, namely CoreSet (CoreSet) and CluMS.

For the large setting, we present budget curves for the datasets with OpenML ID 3 and 1043 in Figure 8. For both datasets, MS is a strong competitor, however CluMS seems to be very strong in the first few iterations. Rand is outperformed by all other strategies, except for the XGB on the first dataset. If the learner achieves high accuracy (as XGB and Catboost do), its probability estimates seem to be reliable and hence information-based strategies are very strong. For the dataset with ID 1043, we observe that CoreSet is initially also quite competitive. If initially the learner has not yet learned too much about the data distribution and achieves also not too good test performance (less than 0.8 accuracy), it might be advantageous to sample representative instances.

In Figure 9, we present budget curves for the datasets with OpenML ID 11 and 51, which both are included in the TabZilla benchmark suite. For the first dataset, one can see that the budget curves for the strong learners RF and TabPFN look quite smooth, especially for TabPFN and also achieve quite high accuracy. The simpler learners k-NN and MLP are struggling more and k-NN even drops in performance in the second half of the active learning procedure. The suitability of different QS again, is quite dependent on the learner: Whereas for the MLP and TabPFN the information-based strategies MS and PowBALD are outperforming the rest, they are the worst when considering k-NN and RF as learners. Regarding the dataset with ID 51, all learners have a hard

time learning the data distribution, as the budget curve is very noisy and also the increases in accuracy are very marginal, except for the MLP. One can deduce, that this dataset definitely is hard for active learning.

In Figure 10, we consider the small setting and present budget curves for the dataset with OpenML ID 334. Overall, the budget curves are much less unstable, compared to the large setting. This is expected, as we start with a very small initial labeled dataset, which makes it really hard to learn the data distribution. The performance of the different QS differs quite a lot for different learners. CoreSet is very strong if the learning algorithms is chosen to be k-NN or TabPFN, whereas for both other learners, the information-based strategies are quite strong. The pipelines consisting of TabPFN as learning algorithm achieve all a much higher accuracy than the pipelines constituted of the other learners. This highlights the importance of choosing an appropriate learning algorithm for the given dataset.

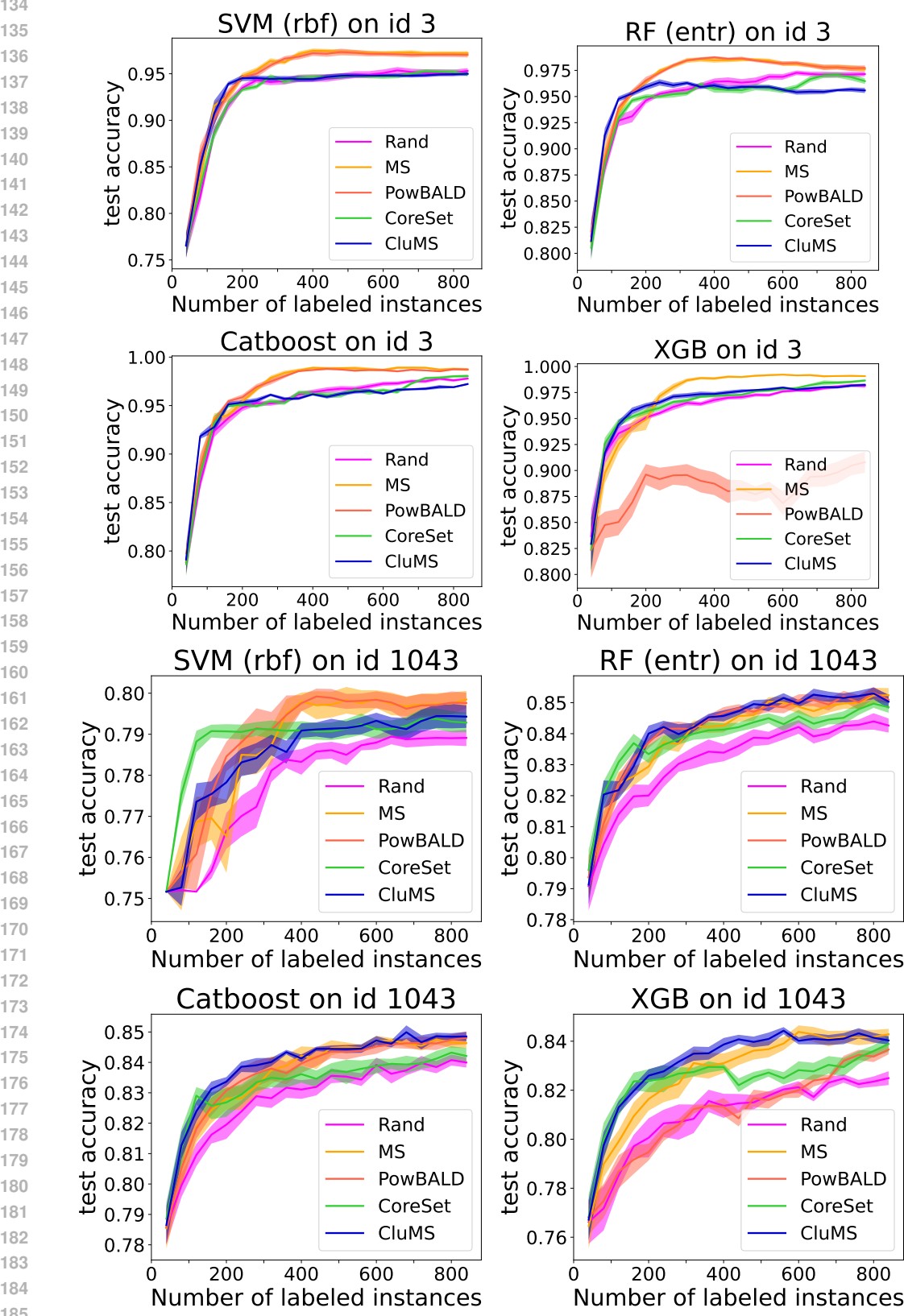

Figure 8: Budget curves for different ALPs on the dataset with OpenML ID 3 and 1043, considering the **large** setting.

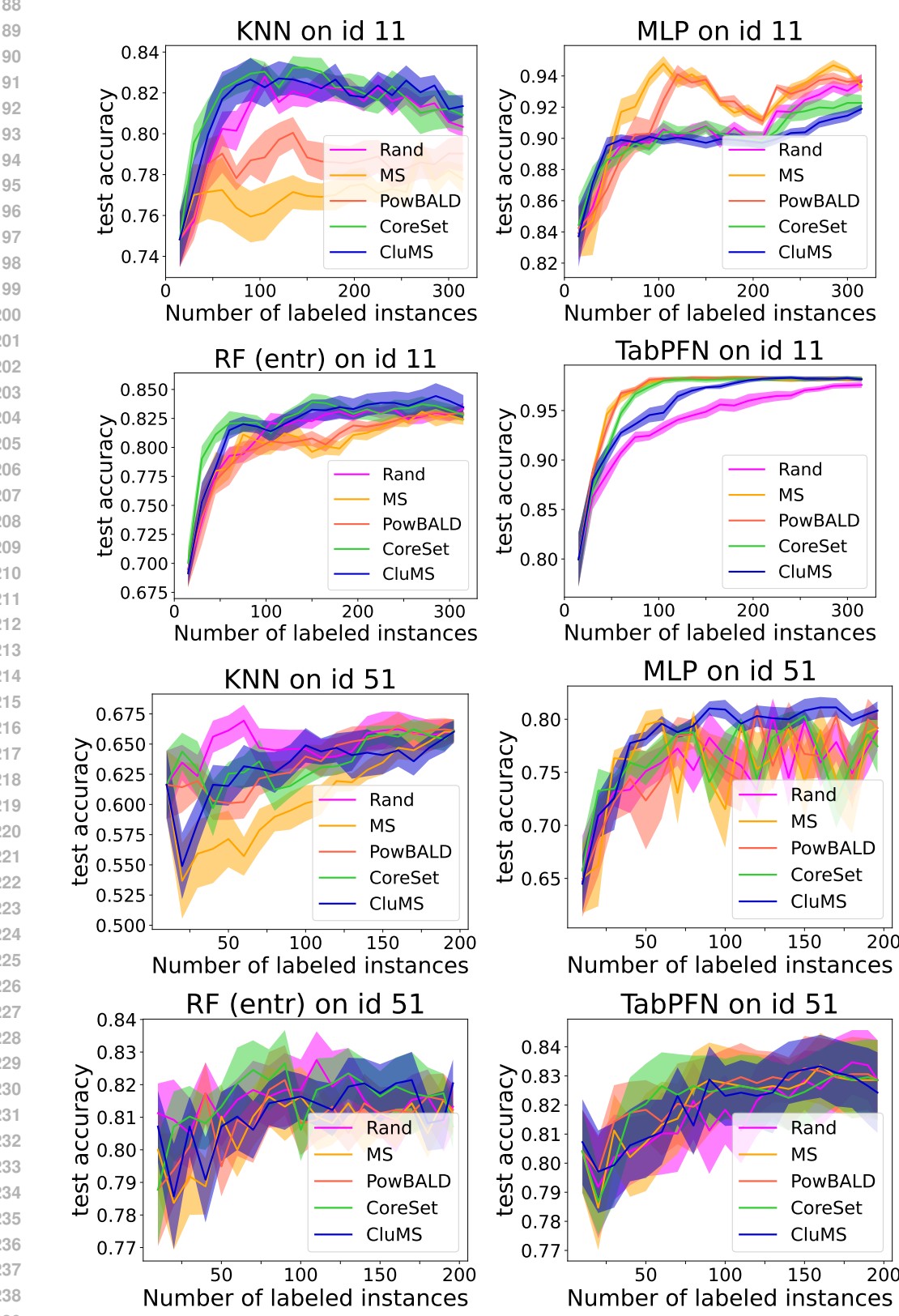

Figure 9: Budget curves for different ALPs on the dataset with OpenML ID 11 and 51, considering the **small** setting.

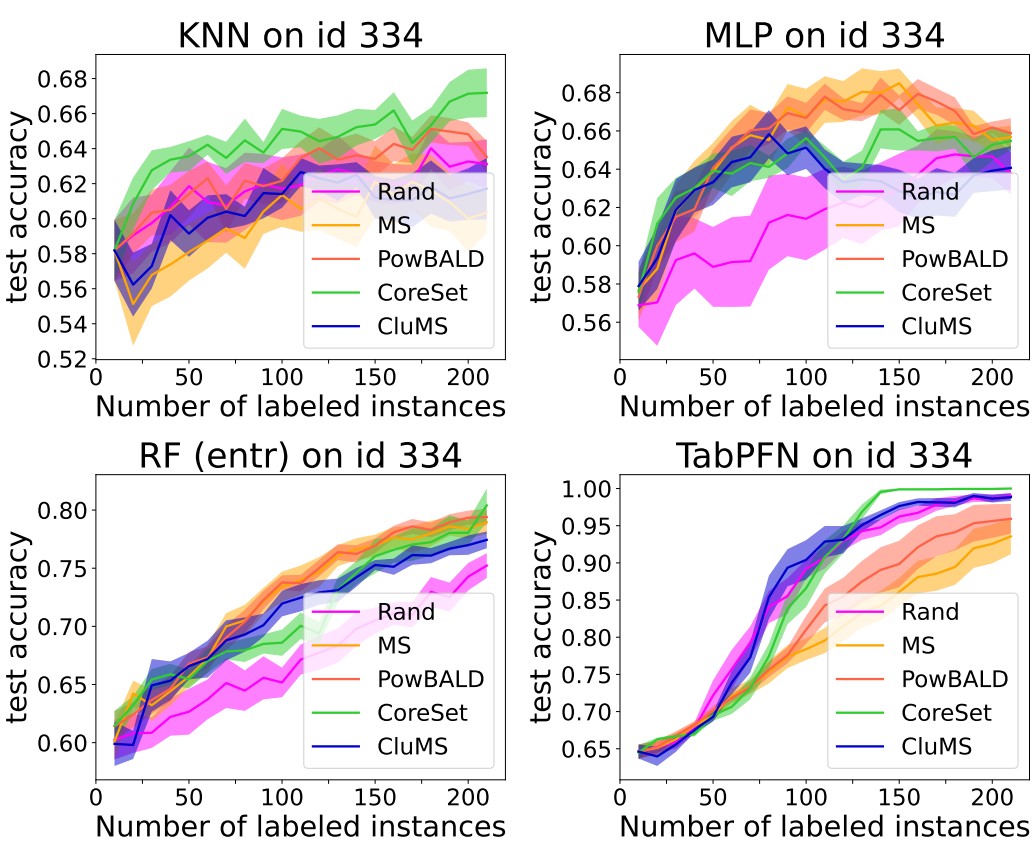

Figure 10: Budget curves for different ALPs on the dataset with OpenML ID 334, considering the **small** setting.

