# OpenReview forum: "ALPBench: A Benchmark for Active Learning Pipelines on Tabular Data"
_ICLR.cc/2025/Conference — Submitted to ICLR 2025_

### Official Review · Reviewer_Dd6v · 2024-10-27

**Soundness:** 3
**Presentation:** 3
**Contribution:** 3
**Rating:** 5
**Confidence:** 3

**Summary:**

This paper proposed a benchmark for active learning on tabular data, in particular, for tabular classification tasks. This aims to fix the issue of the lack of consistent benchmarks for evaluating various active learning methods in various settings with different combinations of learning algorithms and query strategies. The authors also attempt to perform extensive experiments to evaluate the effectiveness of the proposed benchmark.

**Strengths:**

S1: The authors proposed a benchmark to compare different active learning strategies for tabular classification tasks under various settings but in a consistent environment, which is important for performing fair comparisons across the research community.
S2: The benchmark integrates various datasets to compose a comprehensive benchmark, which covers a variety of settings for evaluating active learning strategies

**Weaknesses:**

W1: The scope of this paper is quite narrow, only focusing on tabular classification and only a subset of active learning strategies. However, ICML usually focuses on more general areas including computer vision and NLP. Considering that active learning strategies have been broadly used in those areas, it would be better to take those settings and solutions from those areas into account
W2: I think more learning algorithms should be included, such as the recently emerging transformer model for tabular data, such as the model proposed by "TabTransformer: Tabular Data Modeling Using Contextual Embeddings".
W3: I guess it would be better to discuss some surprising findings by using the proposed benchmark. Although the authors mentioned the decrease in performance in some settings in the experiment section, it would be better to discuss more on that aspect, in particular those ignored by existing studies, thus highlighting the necessity of the proposed benchmark.

**Questions:**

Q1: Can the proposed benchmark be generalized to other modalities and other emerging machine learning models?
Q2: Can the authors discuss more interesting findings that are ignored by existing studies?

---

> ### Author Response · Authors · 2024-11-20
>
> Dear Reviewer Dd6v,
>
> we appreciate your **valuable feedback.** We are happy that you recognized the importance of our benchmark in addressing the lack of consistent frameworks for evaluating active learning strategies. We **refer to our general statement above** and like to address your raised points in the following.
>
> **Weaknesses.**
>
> - W1:
> We are well aware that ICLR traditionally focuses on deep learning and, therefore, primarily on areas such as computer vision and natural language processing. Notably, there is a growing research direction focused on applying **deep learning to tabular data,** particularly for small datasets, leading to, e.g., the recently proposed transformer models like TabPFN, TabNet, or TabTransformer. Thus, we believe exploring active learning within the tabular data domain is an interesting and relevant contribution to this community. Furthermore, we also set the scope of our package explicitly to classification tasks
> on tabular data, as different data domains come with different requirements regarding types of
> learning algorithms and active learning settings. Please also refer to the **"Tabular data" section in the general remark.**
>
>
> - W2: We agree that including additional learning algorithms, such as TabTransformer, would certainly be valuable for expanding the benchmark, and we plan to do it. For this submission, however, our selection of learners was based on [8], aiming to cover a diverse set of algorithms from different model types. In this context, we included TabPFN as a representative of transformer-based models designed for tabular data.
>
>
> - W3: We completely agree that discussing the surprising findings by our benchmark would add significant value to the paper. Therefore, we will add the "Lessons Learned" section to the introduction, as also suggested by Reviewer 3njW.
>
>
>
> **Questions.**
>
> - Q1: ```ALPBench``` can be extended by other machine learning models and to different modalities. Adding custom models is already natively supported, provided they adhere to the scikit-learn API (i.e., implementing the .fit() and .predict_proba() methods). However, extending the benchmark to other data modalities involves additional implementation overhead, as different data modalities require different learning algorithms, preprocessing steps, training procedures, etc.
>
> - Q2: Thank you for this suggestion. We summarized our key findings in the "Lessons Learned" section that we added to the introduction.
>
>
>
> We hope that we have effectively communicated the changes we wish to make to our manuscript
> and addressed any remaining concerns and questions.

---

> > ### Comment · Reviewer_Dd6v · 2024-11-26
> > **Thanks for your comments**
> >
> > I have read the authors' feedback and would like to maintain my current score. But I would love to discuss this with other reviewers if necessary. Thanks!

---

### Official Review · Reviewer_PLPc · 2024-11-01

**Soundness:** 2
**Presentation:** 3
**Contribution:** 2
**Rating:** 5
**Confidence:** 4

**Summary:**

The paper proposes a benchmark for active learning methods
for tabular data. Esp. propose the authors to study active
learning not just for a single down-stream model, but
for a wider selection such as xgboost, catboost, SVMs,
MLPs etc. They suggest to evaluate three different
active learning regimes described by the size of the initial
labeled set, the batch size and the overall budget.
In experiments they compare different active learning
methods with different down-stream models on different
evaluation metrics in different such regimes.

**Strengths:**

- s1. interesting aspect: looking systematically at different down-stream
  models.
- s2. promised a pip installable python module that should be easy to use.
- s3. well written.

**Weaknesses:**

- w1. there is not much innovation in this benchmark besides
  just scaling to more downstream models and more datasets.
- w2. how the three active learning regimes (tab. 1) have been
  chosen is not discussed.
- w3. it is not clearly demonstrated how this new benchmark
  now makes it easier to answer the three research questions
  asked.
- w4. the maybe main question one would want to answer
  by looking at different down-stream models, namely
  should we use different active learning methods for different
  down-stream models, is not addressed.

review.

Looking at different down-stream models that should
be actively learned is clearly useful and maybe often
overlooked in the literature.

However, I see several major issues with the paper
currently:
w1. there is not much innovation in this benchmark besides
  just scaling to more downstream models and more datasets.
- it is not clear which problems the authors had to solve
  to arrive at the current benchmark.
- what are the limitations of current benchmarks, besides
  looking at fewer down-stream models and datasets?

w2. how the three active learning regimes (tab. 1) have been
  chosen is not discussed.
- Esp. it is not clear how these different regimes manage to
  capture similar situations in different datasets. Are not
  30 initial samples for a very simple dataset much, but
  for a more difficult dataset very little?

w3. it is not clearly demonstrated how this new benchmark
  now makes it easier to answer the three research questions
  asked.
- could we not just have used any of the existing benchmarks
  and run it with your 8 downstream models? if not, can you
  describe why not, and how this is now different in your
  benchmark?

w4. the maybe main question one would want to answer
  by looking at different down-stream models, namely
  should we use different active learning methods for different
  down-stream models, is not addressed.
- fig. 2 provides the best active learning pipeline. Not very surprising,
  gradient boosted decision tree models as down-stream models
  are found to perform best on average.
- from an active learning perspective, would it not be more interesting
  to ask the question for the conditional performance of the different
  active learning methods: given a method, say catboost, what are
  the best active learning methods? and esp. are they different
  if I choose different down-stream models, say an SVM instead?

Furter remarks:
- i1. usually the value of a benchmark is seen in identifying an interesting
  set of experiments that allow to ask and answer some research
  questions. You argue your benchmark is large, e.g., but then you
  subsample your own benchmark to provide results. How does this
  provide a clear benchmark for further research in active learning?
- i2. personally, I do not find the heavily colored tables providing
  a very clear overview. A clear decision criterion for what would make
  a good active learning method would be easier to use.


--- added after the rebuttal

Thank you for your replies.

w1. limited innovation
- running the algorithms on more downstream model is "novel" in the sense
  that nobody did it before.
- but there is no difficulty that needs to be solved to do so.
- in this sense, there is limited innovation in the paper.

w2. similar AL regimes
- scaling the number of samples by class, not just absolutely, is standard
  procedure in many areas of ML, e.g., in few-shot-learning.
- besides number of classes, learning tasks vary a lot in difficulty,
  and this is not accounted for in your choice of AL regimes.

w3. no clear demonstration of usefulness of the benchmark.
- "standardized evaluation protocols, pre-defined settings, and logging
  of seeds, exact splits, and model hyperparameters" is done already by
  modern AL benchmarks, e.g., Lüth et al. 2023 and Ji et al. 2023.
  I think you will need another argument for yours.

w4. no  recommendations for per model active learning strategies.
- Yes, you are right, fig. 7 in the appendix I overlooked.

I keep three of my points and slightly update the score accordingly.

additional references:
- Ji, Yilin, Daniel Kaestner, Oliver Wirth, and Christian Wressnegger.
  “Randomness Is the Root of All Evil: More Reliable Evaluation of Deep Active Learning.”
  In Proceedings of the IEEE/CVF Winter Conference on Applications of Computer Vision,
  3943–52, 2023.

**Questions:**

- q1. What is the main problem you had to solve to scale existing
  active learning benchmarks to more downstream models and
  more datasets?
- q2. How did you choose the three active learning regimes in tab. 1?
  How does this capture similar situations in different datasets?
- q3. How did the new benchmark make it easier to answer your three
  research questions? Did you arrive at different conclusion than
  those been drawn in earlier benchmarks?
- q4. What recommendations does your benchmark provide for
  chosing the best active learning method for a down-stream
  learner and based on what evidence?

---

> ### Author Response · Authors · 2024-11-20
>
> Dear Reviewer PLPc,
>
> thanks for your **valuable feedback.** We are pleased that you found our systematic approach to evaluating different downstream models in different settings and metrics interesting and see value in our user-friendly, pip-installable Python package. We **refer to our general statement above** and like to address your raised points in the following.
>
> **Weaknesses.**
>
> - W1: The primary limitation of current active learning benchmarks for tabular data [1-4] is their restriction to a single downstream model. Notably, they consider rather weak models, such as logistic regression, SVMs, and simple MLPs, rather than gradient-boosted trees or more sophisticated deep learning approaches to tabular data. While this may seem like a minor point, we argue that including multiple SOTA downstream models is essential for drawing valid conclusions and have discussed this in more detail in the **general remark in the "Novelty" section**.
>
> - W2: In Appendix A.3, we discuss the criteria behind our choice of settings; however, we will move this information to the main paper for clarity. You are correct that the number of samples required varies with dataset difficulty. This is one reason why we made the number of samples in the dynamic settings (used in our empirical evaluation) dependent on the number of classes, as this measure partially reflects the difficulty of a dataset.
>
> - W3: In principle, it would be possible to use an existing benchmark and adapt it to run with our eight downstream models. However, this would require substantial implementation effort. Also, in contrast to existing benchmarks, our package offers **standardized evaluation protocols, pre-defined settings, and logging of seeds, exact splits, and model hyperparameters.** This is crucial because active learning outcomes can vary widely, as performance is influenced by numerous design choices, including the learner, query strategy, experimental setting, dataset, and evaluation metric.
>
>
> - W4: It is indeed unsurprising that gradient-boosted trees perform strongly as downstream models. However, these models have, to the best of our knowledge, never been evaluated in tabular active learning benchmarks so far. We aimed to address the question of performances of QS conditioned on learners by presenting win-win matrices (Figure 3 and Figure 7 in the Appendix), where we **observed that the suitability of query strategies varies notably across learners**. To clarify and emphasize these findings, we will add a "Lessons Learned" section to the introduction, summarizing these key insights.
>
>
> **Further Remarks.**
> - I1: By subsampling our benchmark, we can provide **more specific and also useful insights**, such as recommending a certain QS for a given learning algorithm and setting, rather than offering broad, general recommendations as identifying the query strategy that performs best on average.
>
> - I2: Our incentive was to summarize results through these tables. However, we acknowledge that a clearer decision criterion could enhance usability, and we have addressed this by **adding the "Lessons Learned" section** in the introduction.
>
> We hope we have effectively communicated the changes we will make to our manuscript and addressed any remaining concerns and questions.

---

### Official Review · Reviewer_3njW · 2024-11-03

**Soundness:** 3
**Presentation:** 3
**Contribution:** 3
**Rating:** 6
**Confidence:** 4

**Summary:**

The paper introduces and illustrates ALPBench, which is a benchmark for reproducible active learning pipelines that covers 86 modern, tabular classification datasets, together with a wide variety of base leaners, query strategies, and setups.

**Strengths:**

The paper fills in a clear gap in today's landscape of active learning for tabular data. The work is reasonably original (for an evaluation framework), clearly presented, and with the potential of having a high-impact in standardizing empirical validations (while also making them apples-2-apples). The wide availability of ALPBench would greatly impact future AL evaluations: far too many of the newly submitted AL papers stop after an arbitrary nmb of queries, w/o any indication on whether or not the achieved performance is in any way meaningful.

**Weaknesses:**

While the paper goes a long way towards standardizing the evaluation of active learners, it can be improved along tow main directions:
1. First of all, instead than the "[somewhat] dry analysis" of the aggregated results in Figures 2 & 3 (which are excellent, but could go into an APPENDIX as supporting evidence), the paper would greatly benefit from a illustrative, step-by-step example of how to use ALPBench in a real-world scenario. Assume that you have a novel tabular dataset NTD for which active learning is essential. Ideally, you would like to follow  a procedure such as (i) identify which base learner BL performs best on NTD - after all, you what to reach SOTA performance with minimum data-annotation cost, (ii) identify which of the existing querying-strategies Q works best with BL for NTD, (iii) identify other datasets have properties similar to NTD, such as having Q+BL as a winner, and (iv) if the results are not satisfactory, invent a novel QS that will deliver better performance that Q+BL, add it to ALPFBench, and re-evaluate. In this scenario, the paper should provide guidance on how to add a new dataset or QS to ALPBench (assuming that it is possible), how to choose between ACC vs AUC metrics, how to choose the best base learner, etc. Similarly, the current insights on what works best for binary vs multi-class, small vs large setup, and the various base learners should be consolidated in a Lessons Leaned section (rather than spread throughout the paper)
2. along the same lines: (i) the paper should provide a table with the SOTA performance on each dataset; that is, which base learner is the best when trained/tuned on all available training/dev data, and (ii) the paper should provide an additional set of metrics that measure how many queries does the best AL approach need to reach 50%, 75%, 90%, 95%, and 99% of the SOTA performance in "(i)". This guarantees that we are doing AL for the right reasons (ie, reach SOTA-adjacent performance with as few labeled examples as possible), rather than measuring "wins" after an arbitrary number of queries

**Questions:**

1. In each row from Table 2, please clarify
    (i) how many of the "capabilities" in the previous four approaches are covered by ALPBench
    (ii) which are missing and why (eg, for QS-Hybr, two of the previous approaches cover 4, while you only cover 3; what is the overlap among them?)

2. line 371: how big are the datasets that you have excluded? given that you have included them in ALPBench, you should -at the very least- give the reader a sense of the challenges/costs/time-constraints/ideal-setup of running experiments at that scale.

3. In how many of the experiments is the performance of the base learner impacted by the 180 secs limit? If this is a real issue, you should have mentioned it at the very beginning of the Experimental Results.

4. is it possible to add to the current QSs two classic strategies such as Query-by-Bagging and Query-by-Boosting? See Abe & Mamitsuka, ICML-1998, "Query Learning Strategies Using Boosting and Bagging."

5. In Fig 4, is KNN the best base learner on those 3 datasets? if not, should we care about the erratic perfornace?

6. Is ALPBench "automatically detecting & reporting" situations in which, after a number of queries, AL hurts rather than helps (as in "5." above")

7. how difficult would it be to extend ALPBench to cover additional scenarios, such as (i) simulating the stream-based scenario, (ii) interleaving AL and semi-supervised learning, and (iii) multiple-view learners?

OTHER:
line 206: please explain intuitively what is epistemic uncertainty; in its current form, it is a bit of a "circular definition:"
              "epistemic uncertainty sampling (EU) (Nguyen et al., 2019) samples instances that have the highest epistemic uncertainty."

---

> ### Author Response · Authors · 2024-11-20
>
> Dear Reviewer 3njW,
>
> thanks for your **valuable feedback** and your **suggestions for improvement.** We also thank you for your appreciation of our work, specifically its potential impact on standardizing future active learning evaluations across different settings using different learners. In the following, we would like to address your proposed opportunities for improvements and answer your questions.
>
> - We will add a "Lessons Learned" section at the end of the Introduction. This section summarizes our insights on the best-performing learner and query strategy combinations for various conditions, such as binary vs. multi-class datasets, small vs. large settings, and AUC vs. Accuracy metric.
>
> - We appreciate the idea of including an illustrative, step-by-step example of using ```ALPBench``` in real-world scenarios, and we will incorporate this. Additionally, we will include a guided example in the Appendix, demonstrating how to add new datasets and query strategies.
>
> - We will also include a table showing the state-of-the-art (SOTA) performance of learners on the full training dataset, as well as use these results to assess how quickly the best active learning approaches reach SOTA performance.
>
> **Questions.**
> - Q1: In Table 3 of Appendix A.2, we present an expanded version of this table, providing a detailed explanation of the query strategies (QS) included in our work and in previous approaches. For our selection of QS, we prioritized widely used, prominent strategies (such as Entropy Sampling, Margin Sampling, and Query-By-Committee) and more recent approaches, e.g. [5,6].
>
> - Q2: In Appendix A.3 (Datasets), we provide details on the excluded datasets and the reasons for their exclusion.
>
> - Q3: A more detailed explanation of the learning algorithm configurations is provided in Appendix A.3. However, we agree that this information would be better placed in the main paper, and we will make this adjustment. Regarding the base learners, it was primarily the TabNet classifier that reached the time limit only in the large setting.
>
> - Q4: Thank you for pointing out that reference. We will add it to Section 3.3 and incorporate it into our benchmark package.
>
> - Q5: For the dataset with ID 846, the Random Forest achieves better performance. For the dataset with ID 1053, KNN does outperform XGBoost; however, this dataset appears to be challenging for Active Learning in general, as performance tends to either decrease or remain stable across both learners (KNN and XGBoost) and all query strategies.
>
> - Q6: No, unfortunately, ```ALPBench``` does not automatically detect or report situations where AL hurts. However, implementing such a feature is challenging, as we typically do not have a validation set available due to the limited amount of labeled data. This question is more closely related to the research area of "early stopping in AL," which is explored, for example, in ["Hitting the target: stopping active learning at the cost-based optimum"](https://link.springer.com/article/10.1007/s10994-022-06253-1).
>
> - Q7: Simulating a stream-based scenario is straightforward - one only needs to define a new setting and set the batch size to 1, which requires minimal effort. Including semi-supervised learning is indeed an interesting idea, and we have considered it; however, we chose to focus exclusively on AL for this submission. However, as we are still developing the package, we definitely plan to incorporate this feature. From an implementation perspective, it would be relatively simple to replace the Oracle with a “SemiSupervisedLabeler.”
>
> - Q8: Epistemic uncertainty refers to the uncertainty arising from a lack of knowledge, and it can be reduced by gathering more data. The argument in the paper is that one should focus on sampling data points with high epistemic uncertainty, as those with high aleatoric uncertainty are unlikely to provide useful information for the learner.
>
>
> We hope that we have effectively communicated the changes we want to make to our manuscript and addressed any remaining concerns and questions.

---

### Author Response · Authors · 2024-11-20
**General Remark**

# General Remark

**We thank the reviewers** for their constructive feedback and valuable suggestions for improvement. We are pleased that our benchmark’s standardized approach to empirical validation is recognized as a **valuable contribution.** However, we noticed that the main novelty of our work - **combining different learning algorithms with query strategies (QS) to evaluate their conditional performances** - may not have been fully conveyed. To address this, we propose a change to the manuscript. We would appreciate an **engagement in discussion** with the reviewers and hope for a raise in scores.

## Changes to the manuscript.

- Add the following **Lessons Learned** section to the end of the introduction to **summarize our key findings**.
```
- Different learners: We confirm that MarginSampling is a highly effective query strategy, particularly when combined with tree-based models. For models like SVM, KNN, and TabNet, representation-based approaches such as
TypicalClustering prove to be better suited. FALCUN performs exceptionally well with MLPs.

- Different datasets: For binary datasets, uncertainty-based methods combined with strong learners prove to be best, as these models provide reliable uncertainty estimates. However, as the number of classes increases, the data
distribution might become more challenging to learn, and the benefit of incorporating representation or
diversity-based approaches becomes more apparent.

- Different settings: The dominance of MarginSampling in the large data setting is reduced in the small setting,
where methods that incorporate diversity excel. In these scenarios, having representative samples becomes more
crucial, whereas in the large data setting, the initial samples may already provide sufficient information to cover
different classes.

- Different metrics: When evaluating for accuracy, Margin Sampling is one of the most effective query strategies. For AUC, methods incorporating diversity, such as, e.g., ClusterMargin or PowerMargin, deliver the best performance.  The importance of diversity might arise because achieving a high AUC requires a well-balanced representation of all
classes in the dataset, ensuring that the model performs equally well across different regions of the decision space.
```


- Provide an **illustrative, step-by-step example** to guide practitioners and researchers on how to apply ```ALPBench``` to their own (real-world) datasets.


## Novelty.
Our work introduces several key innovations that distinguish it from prior efforts.

- **Standardized settings and protocols:** Our benchmark addresses the need for standardization in active learning evaluations by providing **predefined settings and evaluation protocols,** to ensure that **research outcomes are comparable and robust.** In contrast, prior studies typically assess performance in one single (often arbitrary) setting.

- **Reproducibility and experiment tracking:** To ensure reproducibility, we **log exact model parameters and seeds** used in each run. By integrating the Python Package PyExperimenter for standardized experiment logging, we provide a consistent structure for recording results, which is lacking in related work.

- **Native integration of active learning pipelines:** We provide an out-of-the-box setup for combining various query strategies with different learning algorithms. This approach addresses the need for evaluating the **performance of QS conditioned on the chosen learning algorithm** without significant implementation overhead, which prior work often requires.

## Tabular data.

Every benchmark needs to specify its **experimental setup and scope.** Existing benchmarks, such as those for computer vision [1] and also cross-domain learning [8], each come with their own limitations. [1] focuses exclusively on computer vision and evaluates vision transformers as downstream models. [8] spans multiple domains but is restricted to linear models and neural network architectures. In our work, we chose to focus specifically on tabular data.

However, unlike existing tabular benchmarks, we evaluate a diverse range of **state-of-the-art tabular models,** as these are most likely to be used in the downstream task. This includes **tree-based models** (CatBoost and XGBoost) as well as **deep learning models,** such as the transformer-based TabPFN and TabNet.



## References for all rebuttals
[1] Zhang et al., 2024

[2] Bahri et al., 2022

[3] Yang et al., 2016

[4] Zhan et al., 2021

[5] Citovsky et al., 2021

[6] Gilhuber et al., 2024

[7] Ash et al., 2019

[8] McElfresh et al., 2023

[9] Werner et al., 2024

---

### Meta-Review · Area_Chair_VhC4 · 2024-12-10

**Metareview:**

While the reviews agreed that a paper on this very topic is needed to improve the quality of the work on active learning, they also felt that there is still too much room for improvement in the current version and better to have a full revision before publication would be OK. Although this is certainly a disappointment, the message should rather be read as: you are on the right track, keep it up!

**Additional Comments On Reviewer Discussion:**

There was no discussion between authors and reviewers but the reviewers discussed among themselves the pros and cons of the paper with the above mentioned outcome.

---

### Decision · Program_Chairs · 2025-01-22

Reject